# Health Professionals’ Motivational Strategies to Enhance Adherence in the Rehabilitation of People with Lower Limb Fractures: Scoping Review

**DOI:** 10.3390/ijerph20227050

**Published:** 2023-11-10

**Authors:** Júlio Belo Fernandes, Noélia Ferreira, Josefa Domingos, Rui Ferreira, Catarina Amador, Nelson Pardal, Cidália Castro, Aida Simões, Sónia Fernandes, Catarina Bernardes, Diana Alves Vareta, Dina Peças, Dora Ladislau, Natacha Sousa, Andreia Duarte, Catarina Godinho

**Affiliations:** 1Egas Moniz Center for Interdisciplinary Research (CiiEM), Egas Moniz School of Health & Science, 2829-511 Almada, Portugal; nferreira@egasmoniz.edu.pt (N.F.); domingosjosefa@gmail.com (J.D.); cbernardes@egasmoniz.edu.pt (C.B.); diana_vareta@hotmail.com (D.A.V.); dinabaiao@sapo.pt (D.P.); cgcgodinho@gmail.com (C.G.); 2Nurs* Lab, 2829-511 Almada, Portugal; 3Department of Nursing, Hospital Garcia de Orta, 2805-267 Almada, Portugalcatarina.amador@hgo.min-saude.pt (C.A.);; 4Regional Health Administration of Lisbon and Tagus Valley, Community Care Unit-Integrating Health, 1500-534 Lisbon, Portugal; 5Department of Nursing, Setúbal Hospital Centre, 2900-182 Setúbal, Portugal

**Keywords:** fractures, bone, hip fractures, motivational interviewing, patient compliance, rehabilitation

## Abstract

Patients with lower limb fractures require rehabilitation but often struggle with adherence to interventions. Adding motivational strategies to rehabilitation programs can increase patient adherence and enhance outcomes. This review aims to identify the motivational strategies used by health professionals in the rehabilitation of people with lower limb fractures. We used Arksey and O’Malley’s methodological framework to structure and conduct this scoping review. The literature search was performed using the Scopus, CINAHL, MEDLINE, Nursing & Allied Health, and Cochrane Central Register of Controlled Trials databases. The final search was conducted in February 2023. A total of 1339 articles were identified. After selecting and analyzing the articles, twelve studies were included in this review. Health professionals use several strategies to motivate patients with lower limb fractures to adhere to rehabilitation programs. These strategies include building a therapeutic alliance, increasing patients’ health literacy, setting achievable goals, personalizing the rehabilitation program, managing unpleasant sensations of exercise, using persuasion, providing positive reinforcement, avoiding negative emotional stimulation, and helping to seek support. The motivational strategies identified may help professionals to increase patient adherence to rehabilitation for lower limb fractures. This knowledge will allow these professionals to help patients overcome barriers to rehabilitation, enhance their motivation, and ultimately improve their recovery outcomes.

## 1. Introduction

A fracture is a medical condition that disrupts the continuity of bone, either as a complete or incomplete break caused by high-force impact, stress, a minor injury, or certain medical conditions [1]. The type of fracture and the location on the body can differ significantly based on various factors, including the quality of an individual’s bones and the nature of the trauma experienced [2]. Fractures are a significant public health issue that can occur in people in all age groups resulting in a range of adverse outcomes, including work absenteeism, decreased productivity, disability, and health complications, ultimately affecting the patient’s quality of life and creating high healthcare expenses, thus imposing a significant burden on individuals, families, societies, and healthcare systems [3,4].

The incidence of lower limb fractures worldwide varies depending on several factors, including age, sex, socioeconomic status, and geographic location. While comprehensive data on the incidence of lower limb fractures is not available for all countries, some general trends can be observed [3].

Globally, in 2019, there were over 178 million new fractures; the fracture of the patella, tibia, fibula, or ankle was the most common type of fracture, with an estimated 32.7 million cases. In addition, lower limb fracture incidence reached 72.2 million [3].

Physical therapy is a widely acknowledged therapeutic approach in patients with lower limb fractures [5]. In addition, health professionals often recommend home exercise programs to patients as part of their clinical rehabilitation or to manage long-term health conditions independently [6].

Following the recommended rehabilitation program and maintaining adherence can lead to long-term health benefits, such as improved physical function and reduced pain [7]. It can also significantly enhance a patient’s quality of life and alleviate the burden on healthcare systems by allowing patients to manage their health independently [8]. However, a well-documented challenge with rehabilitation interventions is the limited adherence to home exercise programs [9].

Low levels of adherence can curtail the efficacy of rehabilitation and potentially result in the recurrence of injuries or the inability to self-manage chronic conditions, ultimately leading to diminished functionality and adverse outcomes [10].

In patients with lower limb fractures, the development of strategies to motivate patients to adhere to a rehabilitation program is fundamental, as there is evidence that many older people with lower limb fractures worry about their future ability to walk again and perceive the fracture to be the end of independent life [11,12,13].

Previous research has shown that exercise self-efficacy positively correlates with initiating and maintaining physical exercise, particularly in a preplanned program’s early and middle stages [14,15,16,17].

Bandura [18] defines self-efficacy as the individual’s belief in their ability to achieve a specific goal or perform a particular task. At the same time, motivation refers to the drive or desire to perform that behavior. According to Bandura, self-efficacy is crucial in determining an individual’s motivation, behavior, and performance [18]. Several factors can influence both self-efficacy and motivation. Bandura’s theory proposes four sources of self-efficacy: mastery experiences, vicarious experiences, verbal persuasion, and physiological and affective states. Mastery experiences involve completing a task or achieving a goal, which increases an individual’s belief in their ability to perform similar tasks in the future. Vicarious experiences involve observing others like oneself successfully performing a task, which can increase one’s self-efficacy. Social persuasion consists of receiving positive feedback or encouragement from others, increasing one’s belief in their ability to perform a task. Finally, physiological and affective states refer to an individual’s physical and emotional reactions to a task, which can either increase or decrease their belief in their ability to perform it [18,19].

In various fields, including education, sports, and healthcare, Bandura’s self-efficacy theory has been applied to help individuals improve their motivation, behavior, and performance. Although there are several reasons why a patient might struggle to adhere to a rehabilitation program, previous studies have reported that interventions that increase motivation to adhere to rehabilitation programs can be effective [20,21,22]. Therefore, with this review, we aim to identify the motivational strategies used by health professionals in the rehabilitation of people with lower limb fractures.

## 2. Materials and Methods

This study employed a scoping review following the five stages outlined in the Arksey and O’Malley methodological framework [23], namely: Stage 1—identifying the research question; Stage 2—identifying relevant studies; Stage 3—study selection; Stage 4—charting the data; and Stage 5—collating, summarizing, and reporting the results. The protocol was registered with the Open Science Framework (https://doi.org/10.17605/OSF.IO/ZPU36, accessed on 16 April 2023). In addition, the report was drawn based on the Preferred Reporting Items for Systematic Reviews and Meta-Analyses extension for Scoping Reviews (PRISMA-ScR) checklist [24].

### 2.1. Stage 1: Identifying Research Questions

The first step was to formulate a clear and specific research question to steer the scoping review process. The research question was generated by the PCC framework (population, concept, and context). This framework is commonly used in scientific reviews to help researchers precisely define the fundamental elements that shape their study’s scope and focus. Researchers using the PCC framework ensure that their research question, objectives, and inclusion/exclusion criteria are meticulously defined and aligned with the unique aspects of the study, including the specific population, concept, and context [25].

To guide this scoping review, we formulated the following research question: what are the motivational strategies (C) used by health professionals in the rehabilitation (C) of people with lower limb fractures (P)?

### 2.2. Stage 2: Identifying Relevant Studies

The literature search was conducted using the Scopus, CINAHL, MEDLINE, Nursing & Allied Health, and Cochrane Central Register of Controlled Trials electronic databases. The final search was conducted on 22 February 2023.

Both medical subject headings and free-text search terms were employed with Boolean operators to construct the following search string:

((Hip Fractures) OR (Femoral Fractures) OR (Arthroplasty, Replacement, Hip) OR (Knee Fractures) OR (Ankle Fractures) OR (Tibial Fractures) OR (Fibula Fractures) OR (Arthroplasty, Replacement, Ankle) OR (Arthroplasty, Replacement, Knee)) AND ((Motivation) OR (Adherence) OR (Patient Compliance)) AND ((Rehabilitation) OR (Exercise) OR (Physical Therapy Modalities) OR (Physiotherapy) OR (Physical therapy)).

The PCC framework laid the groundwork for shaping the search strategy and defining the criteria for inclusion and exclusion (Table 1).

The review included qualitative and quantitative studies focusing on strategies health professionals use to motivate people with lower limb fractures. Approved studies had to be written in English, Portuguese, or Spanish and published between 2003 and 2023.

We selected this time frame to balance research depth and breadth, allowing us to include contemporary and relevant studies in the field while considering practical limitations related to resources and time. This approach enables a comprehensive analysis that effectively represents the diversity of motivational strategies.

All documents that did not meet the selection criteria were excluded from this scoping review.

### 2.3. Stage 3: Study Selection

After removing duplicates, the citation titles and abstracts of each study were screened independently by two researchers. Relevant studies were then obtained in full text, and the researchers evaluated each study independently to ensure complete consensus on whether it met the inclusion/exclusion criteria. In case of any disagreements over eligibility, the researchers discussed the matter with the additional researchers to reach a consensus. The inter-rater reliability at the end of the screening process reached 94.7%, and it was 100% for the eligibility process.

### 2.4. Stage 4: Charting the Data

Two reviewers performed data extraction using a designed instrument tailored to meet the specific requirements of the research question guiding the review. This instrument was structured to ensure a comprehensive collection of relevant data [23].

The reviewers extracted a range of data items, including general information such as the author’s name, the year of publication, the title of the study, and the country where the research was conducted. These general data are crucial for establishing the context and background of the studies under review.

Methodological data were another essential component of the extraction process. This category encompassed details related to the study’s design and aim. Lastly, the reviewers extracted data about the results of the studies, mainly focusing on the strategies used to motivate participants.

All authors participated in a comprehensive review and discussion of the final extraction chart. This collaborative approach ensures that the extracted data are accurate and relevant, contributing to the overall quality and validity of the scoping review.

### 2.5. Stage 5: Collating, Summarizing, and Reporting the Results

The PRISMA flow diagram shows the studies’ identification, screening, and selection (Figure 1).

A data-driven thematic analysis was adopted from Braun, Clarke, Hayfield, and Terry’s guidelines [26] to organize and summarize data. Two researchers independently reviewed the data, manually coding them through analysis to identify common themes that emerged from the collected data based on the four sources of self-efficacy proposed by Bandura [18,19].

## 3. Results

This scoping review identified several strategies health professionals used to motivate patients with lower limb fractures to adhere to rehabilitation programs. These strategies were highlighted and transferred into Table 2, which provides an overview of their key characteristics and findings.

Out of the twelve studies, there were seven studies conducted in Europe (United Kingdom [27,28], Denmark [29], Finland [30], Norway [31], Sweden [32], and the Netherlands [33]), two in the United States of America [34,35], two in China [36,37], and another in South Korea [38].

Of the twelve relevant studies, seven were randomized controlled trials [27,29,30,33,35,37,38], two were protocols for a randomized controlled trial [28,36], one was a quasi-experimental prospective study [32], one was a secondary data analysis from a randomized controlled trial [34], and one was a qualitative descriptive study [31].
ijerph-20-07050-t002_Table 2Table 2Data extraction and synthesis.Author/Year/Title/CountryStudy Design/AimInterventionsOlsson et al., 2007 [32]Effects of nursing interventions within an integrated care pathway for patients with hip fracture SwedenQuasi-experimental prospective study. To evaluate the contribution of nursing care within an integrated care pathway for patients with hip fractures.A thorough interview on admissionGoal settingFamily involvementOral and written information about the care programResnick et al., 2007 [35]Testing the effectiveness of the exercise plus program in older women post-hip fractureUnited States of AmericaRandomized controlled trial.To test the impact of a self-efficacy-based intervention, the Exercise Plus Program, and the different components of the intervention on self-efficacy, outcome expectations, and exercise behavior among older women post-hip fracture.Education about the benefits of exerciseVerbal encouragementGoal settingPositive reinforcementExposure to interventions to decrease the unpleasant sensations associated with exerciseCueing with posters describing the exercisesCalendar of daily exercise activitiesCasado et al., 2009 [34]Social support for exercise by experts in older women post-hip fractureUnited States of AmericaSecondary data analysis from a randomized controlled trial.To examine how social support for exercise by experts affected the self-efficacy, outcome expectations, and exercise behavior among older women following a hip fracture.Education about the benefits of exerciseVerbal encouragementGoal settingPositive reinforcementExposure to interventions to decrease the unpleasant sensations associated with exerciseCueing with posters describing the exercisesCalendar of daily exercise activitiesSipilä et al., 2011 [30]Promoting mobility after hip fracture (ProMo): study protocol and selected baseline results of a year-long randomized controlled trial among community-dwelling older peopleFinlandRandomized controlled trial.To describe the design, intervention, and demographic baseline results of a study investigating the effects of a rehabilitation program aiming to restore mobility and functional capacity among community-dwelling participants after hip fracture.Individually tailored exercise programMotivational face-to-face counseling sessionThe problem-solving method was used to address perceived obstacles to physical activity and to access exercise facilitiesWritten informationPhone contactsPhysical activity diaryHarmelink et al., 2017 [33]The effectiveness of the use of a digital activity coaching system in addition to a two-week home-based exercise program in patients after total knee arthroplasty: study protocol for a randomized controlled trialthe NetherlandsRandomized controlled trial.To determine the effectiveness of an activity coaching system in addition to a home-based exercise program after a TKA compared to only the home-based exercise program with physical functioning as the outcome.Digital activity coaching systemRoom et al., 2020 [28]Development of a functional rehabilitation intervention for post-knee arthroplasty patients: Community based Rehabilitation post-Knee Arthroplasty (CORKA) trialUnited KingdomStudy protocol for a randomized controlled trial.To develop a functional rehabilitation intervention for the CORKA trial.Individually tailored exercise programEducationInformation bookletGoal settingExercise diaryBehavioral contractVestøl et al., 2020 [31]The importance of a good therapeutic alliance in promoting exercise motivation in a group of older Norwegians in the subacute phase of hip fracture; a qualitative studyNorwayQualitative descriptive study.To explore how older people who had participated in an evidence-based exercise intervention describe their relationship with their therapists and how this relationship might contribute to their motivation for exercise.Feeling of mutuality and respect in the allianceA trusting and motivating relationshipTailoring the instruction and program to make the task understandableBarker et al., 2021 [27]Home-based rehabilitation programme compared with traditional physiotherapy for patients at risk of poor outcome after knee arthroplasty: the CORKA randomised controlled trialUnited KingdomRandomized controlled trial.To evaluate whether a home-based rehabilitation program for people at risk of a poor outcome after knee arthroplasty offers superior outcomes to traditional outpatient physiotherapy.Individually tailored exercise programEducationInformation bookletGoal settingExercise diaryBehavioral contractMeng et al., 2022 [37]Effectiveness of self-efficacy-enhancing interventions on rehabilitation following total hip replacement: a randomized controlled trial with six-month follow-upChinaRandomized controlled trial.To explore the effect of a self-efficacy-enhancing intervention program following hip replacement on patients’ rehabilitation outcomes (self-efficacy, functional exercise compliance, hip function, activity and social participation, anxiety and depression, and quality of life).Providing knowledge of functional exerciseSetting achievable goalsProviding positive feedbackSharing cases of successful rehabilitationPersuasionGiving verbal encouragement and complimentsAvoiding negative emotional stimulationHelping participants seek social supportBieler et al., 2022 [29]Effectiveness of promotion and support for physical activity maintenance post total hip arthroplasty-study protocol for a pragmatic, assessor-blinded, randomized controlled trialDenmarkRandomized controlled trial.To investigate whether the promotion and support of physical activity initiated 3 months after total hip arthroplasty complementary to usual rehabilitation care can increase objective measured physical activity 6 months post-surgery.Motivation interviewingInformation leafletGoal settingTelephone-assisted counselingDeng et al., 2022 [36]A self-efficacy-enhancing intervention for Chinese patients after total hip arthroplasty: study protocol for a randomized controlled trial with 6-month follow-up ChinaStudy protocol for a randomized controlled trial.To develop and assess the feasibility of a self-efficacy-enhancing intervention to improve exercise adherence in patients undergoing total hip arthroplasty.Setting achievable goalsProviding information on the benefits of functional exerciseProviding positive feedbackSharing cases of successful RehabilitationVerbal encouragement, explanation, and persuasionDeveloping strategies to cope with barriersHelp to seek social supportLee and Lee, 2022 [38]Effectiveness of Multicomponent Home-Based Rehabilitation in Elderly Patients after Hip Fracture Surgery: A Randomized Controlled Trial South KoreaRandomized controlled trial.To assess the clinical effectiveness of an 8-week personalized multicomponent home-based rehabilitation program by comparing it with a home exercise program after discharge.Individually tailored exercise programEducationInformation leafletMotivational counseling

As presented in Table 1, there are several interventions used by health professionals to motivate patients with lower limb fractures to adhere to a rehabilitation program. These interventions were incorporated into the four major sources of influence on self-efficacy defined by Bandura [18,19] (Table 3) and detailed below.

### 3.1. Mastery Experiences

According to Bandura [19], mastery experiences result from a person’s direct experiences of success or failure in a particular task or situation. This past performance can be a source of effective influence for developing self-efficacy. In this domain, we have identified several studies that used strategies such as building a therapeutic alliance [30,31,32], increasing health literacy [24,27,28,29,32,34,35,36,37,38], setting achievable goals [27,28,29,30,32,35,36,37], personalizing the rehabilitation program [27,28,30,31,38], and managing unpleasant sensations [34,35]. To implement these strategies, health professionals apply specific techniques to develop a trusting and motivating relationship based on feelings of mutuality and respect in the alliance [31], aiming to know the individual’s personhood and identify their needs. This knowledge enables health professionals to help patients set achievable goals at different stages of the rehabilitation process [27,28,29,32,34,35,36,37] and tailor the exercise program to their needs and abilities [27,28,30,38], motivating patients to engage in regular physical activities. Another specific technique used to motivate patients is increasing their knowledge regarding the care program and functional exercise and its benefits [34,35,37]. Several studies have used leaflets or booklets to complement oral information [24,27,28,29,32,38]. To manage the unpleasant sensations associated with exercise, only one study describes the use of prescribed medications or heat/ice treatment to relieve or decrease pain [34,35]. In addition, encouraging patients to observe and maintain a physical activity diary [27,28,30] will record their exercise behavior, enabling them to track key metrics, determine what works in the exercise routine, and progress toward the set goals.

### 3.2. Vicarious Experience

According to Bandura’s self-efficacy model, vicarious experience involves observing other people’s experiences and successes in similar tasks or situations. By observing the success of others in similar tasks or situations, individuals can develop a sense of confidence and motivation that can help them achieve their goals [19]. In this domain, we have identified sharing cases of successful rehabilitation [36,37] and the use of the problem-solving method [30]. Researchers shared the successful experiences of other patients to build confidence and used the problem-solving method to address perceived obstacles to physical activity and access to exercise facilities. In addition, by introducing previous success stories, researchers aimed to motivate participants to adhere to the rehabilitation program in the following month.

### 3.3. Verbal Persuasion

In Bandura’s self-efficacy model, verbal persuasion is one of the four primary sources of information that can influence an individual’s self-efficacy beliefs. Verbal persuasion refers to the feedback and encouragement an individual receives from others. Verbal persuasion can be a powerful tool in shaping an individual’s self-efficacy beliefs. It can provide them with positive reinforcement, constructive feedback, and practical advice, affecting how they feel about their capability to cope with the challenge [19]. In this domain, we have identified two strategies: persuasion [27,28,29,37] and giving verbal encouragement and compliments [29,30,32,33,34,35,36,37,38]. Researchers appealed to specific techniques such as describing the benefits of physical activities [37], asserting that participants can self-manage [37], providing positive verbal feedback upon their efforts, and giving verbal encouragement and compliments [34,35,36,37]. Researchers also maintained regular phone contact with patients to reinforce participants’ past and present successes or accomplishments [29]. In addition to encouraging family involvement to provide additional positive reinforcement and encouragement [32], one study used a digital activity coaching system [33]. This digital system provided time-based motivational cues in the form of messages on a smartphone during the day to create awareness concerning how active the patient had been up until that moment and extra motivation.

### 3.4. Physiological and Affective States

Bandura also suggests that an individual’s self-efficacy beliefs can influence physiological states. For example, people with high self-efficacy in a particular task or situation may experience physiological responses known as arousal that can help prepare the individual for action. In contrast, someone with low self-efficacy may experience physiological responses that can hinder their performance [19]. Researchers used two strategies in this domain to motivate participants: Avoid negative emotional stimulation [37] and help participants seek support [29,30,36,37]. To develop these strategies, researchers assess patients’ expressions of anxiety and depression [37], identify individual barriers and resources for performing the exercise plan [30,37], and provide strategies for dealing with the identified barriers and coping in the future [30,36,37].

## 4. Discussion

This review identified several strategies health professionals used to motivate patients with lower limb fractures to adhere to a rehabilitation program.

Bandura’s self-efficacy model explains that an individual’s belief in their ability to perform a specific task or achieve a particular goal can influence their behavior, thoughts, and emotions [19]. The beliefs become a primary, explicit explanation for motivation, as individuals are more likely to engage in behaviors if they believe they can succeed; therefore, in rehabilitation following a lower limb fracture, an individual’s self-efficacy can impact their adherence to their rehabilitation program.

Several studies focused on self-efficacy as the target of the intervention. Researchers used strategies concentrated on education and follow-ups to enhance self-efficacy from its four main domains (mastery experiences, vicarious experience, verbal persuasion, and psychological monitoring).

Health professionals can enhance patients’ adherence to rehabilitation plans by increasing self-efficacy through intervention strategies based on Bandura’s theory. The mastery experiences included building a therapeutic alliance that fosters a sense of reciprocity and respect in the alliance. The therapeutic alliance is crucial for successful rehabilitation as it fosters a safe and supportive environment for patients to explore their emotions and experiences [39]. It helps establish a rapport and understanding between the health professional and the patient, which is crucial for supporting them. At the same time, they are encouraged to take an active role in their healing process [39,40]. Research has consistently shown that a solid therapeutic alliance is associated with positive health outcomes, including increased satisfaction, improved symptoms, and better long-term results [41,42,43]. This trusting and motivating relationship enables the identification of the individual’s previous experiences and the challenges of self-management needs through discussions that serve as a basis for setting achievable goals and actions to improve self-management [44,45].

Health professionals can tailor the rehabilitation program specifically to meet individual needs, goals, and abilities based on the knowledge acquired about the patient. Evidence shows that tailored rehabilitation programs effectively enhance patient adherence [46,47,48,49,50]. Therefore, creating an exercise plan customized to the patient’s needs, goals, and abilities is crucial while considering their medical conditions.

While exercise is essential for patients with lower limb fractures to obtain positive health outcomes, there is an associated risk of developing unpleasant sensations associated with exercise, such as pain. To manage these sensations, health professionals can resort to adjusting the exercise plan, prescribed medications, or heat/ice treatment [35].

Another strategy used to enhance self-efficacy was maintaining a physical activity diary. This record allows patients to self-monitor their daily progress, objectively review their actions and emotions toward their goals, and adjust their behaviors accordingly [51]. A previous study also showed that keeping an activity diary enhanced therapy outcomes, decreased pain levels, and prevented the deterioration of patients’ physical abilities [52].

Finally, through health promotion, health professionals can enhance individuals’ lifestyle behaviors and their knowledge and attitudes toward rehabilitation programs. Previous studies have shown that increased knowledge and attitudes are significant forerunners of behavior changes [53].

Vicarious experiences include sharing examples of people who had experienced comparable experiences and successfully used self-management skills to decrease their distress. Combining this strategy with the problem-solving method, health professionals can enhance patients’ abilities to address perceived barriers to physical activity and outline strategies that enable them to overcome these barriers.

Vicarious experiences can be a valuable tool in rehabilitation, as they allow individuals to learn and acquire new skills by observing and learning from others. This approach can be particularly effective for individuals with difficulty learning through more traditional methods, such as verbal instruction or hands-on practice [54]. By observing others who have successfully managed their symptoms, individuals can gain confidence in their ability to do the same. In addition to providing a means of learning new skills, vicarious experiences can also help to build social connections and a sense of community among individuals in rehabilitation [55]. By seeing others going through similar challenges and working toward similar goals, individuals can feel less isolated and more supported in their efforts to recover and regain independence.

Health professionals can use verbal persuasion as a tool to motivate their patients. Verbal persuasion comprises encouragement and acknowledging the participants’ abilities to perform a specific task. By describing the benefits of performing exercise and asserting that patients can self-manage, health professionals can provide positive feedback for the patient’s effort, give verbal encouragement, and reinforce their past and present successes. Family involvement in verbal persuasion and using a digital support system can act as extra support and motivation. In addition to helping strengthen the patients’ self-efficacy by providing positive reinforcement, verbal persuasion can create further opportunities for communication, support, and education to enhance patients’ self-efficacy. Importantly, there is an increasing drive to incorporate digital interventions into rehabilitation practices [56]. Digital support systems are comparatively simple and cost-effective, subject to patients’ literacy and physical ability. In the face of the growing pressure on physical rehabilitation services, the addition of interventions that combine digital with face-to-face treatments can be a path to promote effective clinical management and increase the self-management of conditions [56,57].

Physiological and affective states should allow the assessment of stress expressions and help patients seek support by discussing strategies for managing the identified challenges. These interventions can be critical for the person’s successful rehabilitation because many older adults see a hip fracture as the end of independent living [13]. Therefore, psychological monitoring may be essential to encourage patients and show them that recovery is possible. Helping patients identify strategies to manage challenges is a crucial aspect of healthcare. It can improve the patient’s overall well-being, enhance their quality of life, and support them in achieving their health goals [58].

Patients undergoing lower limb fracture rehabilitation must receive functional training and physical activity, as the fracture can significantly impact a person’s mobility, strength, and balance. Therefore, they need to adhere to a rehabilitation program. Strategies to promote adherence should be an essential focus for those who prescribe exercise [59]. As the rehabilitation workforce has experience caring for patients with lower limb fracture, they are aware of and understand the challenges these patients might face during treatment. Further, as they work in direct contact with the patients, they build therapeutic alliances reinforcing adherence to the rehabilitation plan. Therefore, providing additional training to prepare these health professionals to promote self-efficacy-enhancing intervention is essential and not resource-intensive.

### Strengths and Limitations

This review has merits by identifying different strategies to motivate patients with lower limb fractures to adhere to rehabilitation programs.

In addition, to serve as a platform for helping other professionals to motivate patients, the findings from this review can have implications for various stakeholders.

The diverse strategies uncovered in this review lay the groundwork for future research endeavors. They provide a rich resource for researchers to delve deeper into the effectiveness and nuances of these motivational techniques. By building upon the foundation established in this review, researchers can explore innovative approaches, assess long-term outcomes, and refine existing strategies.

The strategies unveiled can readily translate into practical interventions for healthcare professionals. The review is a resource to guide health professionals in their daily practice. It equips them with a toolbox of effective motivational techniques that they can integrate into their patient care routines. This, in turn, can enhance patient engagement and adherence, ultimately improving the quality of care.

One of the notable implications is the potential development of specialized care pathways. The findings can serve as a foundational structure to design a care pathway explicitly aimed at motivating patients with lower limb fractures to adhere to rehabilitation programs. Such a pathway can outline a systematic and evidence-based approach to treatment, addressing the unique needs and challenges of this patient population.

The findings of this review not only provide immediate value but also have far-reaching implications. They serve as a foundation for future research, inspire the development of tailored care pathways, empower healthcare professionals, and, most importantly, contribute to the well-being and recovery of patients with lower limb fractures. Nonetheless, this research has several limitations. First, although we have identified different strategies, the information needs more detail. For example, there needed to be more information regarding the booklets or leaflets’ contents or how to tailor the exercise program to each patient’s needs and abilities, making capturing the essence of interventions more challenging. Second, limiting the search to five databases may have excluded some relevant studies. Additionally, the selected descriptors for the search may also have excluded pertinent articles. Third, only English, Portuguese, and Spanish papers were considered for this review. Therefore, we must consider the possibility that there are studies published in other languages that would have changed the review results. Finally, adhering to the framework proposed by Arksey and O’Malley, the review does not incorporate the assessment of study quality as part of the scoping process.

A growing body of evidence focuses on interventions for patients with lower limb fractures; however, this review shows a gap regarding the strategies for motivating them to adhere to rehabilitation programs, as only twelve studies were identified. Further research is needed based on the possible benefits of using motivational strategies for patients with lower limb fractures to adhere to rehabilitation programs and the scarcity of studies focusing on this subject.

## 5. Conclusions

This scoping review identified several strategies health professionals use to motivate patients with lower limb fractures to adhere to a rehabilitation program. Lower limb fractures can significantly impact patients’ physical and mental health, and successful program fulfillment can be critical for their recovery. However, many patients need help with adherence to these programs for various reasons. Therefore, health professionals must employ motivational strategies to encourage patients to adhere. These strategies may include building a therapeutic alliance, increasing patients’ health literacy, setting achievable goals, personalizing the rehabilitation program, managing unpleasant sensations of exercise, using persuasion and providing positive reinforcement, avoiding negative emotional stimulation, and helping to seek support. By implementing these strategies, health professionals can empower patients to overcome the barriers to rehabilitation, enhance their motivation, and ultimately improve their recovery outcomes.

## Figures and Tables

**Figure 1 ijerph-20-07050-f001:**
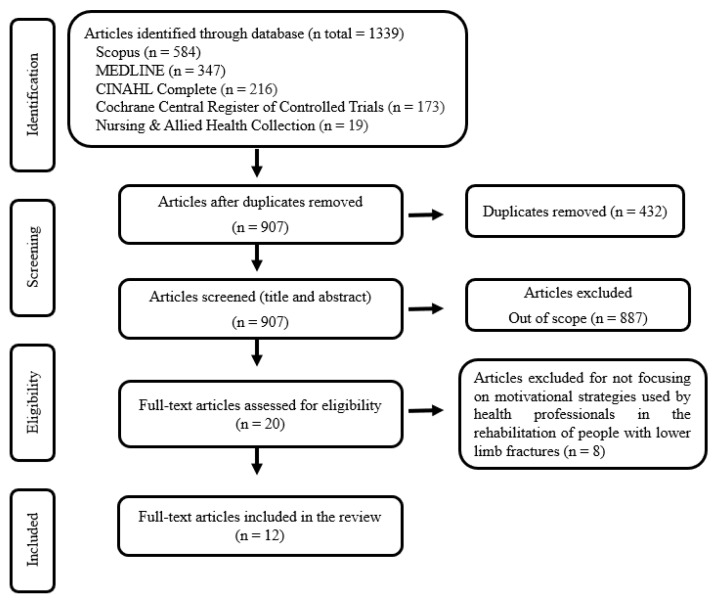
PRISMA flow diagram for study selection.

**Table 1 ijerph-20-07050-t001:** Eligibility criteria.

Parameter	Inclusion Criteria	Exclusion Criteria
Population	Patients with lower limb fractures; Adults ≥ 18 years old.	Other health conditions besides lower limb fractures; Participants < 18 years old.
Concept	Studies that explore motivational strategies developed by healthcare professionals.	Studies that do not address motivational strategies.
Context	Studies conducted in rehabilitation settings (e.g., acute, post-acute, and long-term care).	Studies conducted in non-healthcare or non-rehabilitation settings.

**Table 3 ijerph-20-07050-t003:** Strategies and interventions.

Domain	Strategies	Interventions
Mastery experiences	Therapeutic alliance	A thorough interview on admission.
		Develop a trusting and motivating relationship.
		Feelings of mutuality and respect in the alliance.
		Face-to-face counseling sessions.
	Health literacy	Educate patients in rehabilitation exercise, complications, disease, and the benefits of exercise.
		Provide information leaflets/booklets.
		Cueing with posters describing the exercises.
	Set achievable goals	Identify patients’ abilities and needs.
		Confer with patients to develop functional exercise goals at different stages of rehabilitation.
		Physical activity diary.
		Calendar of daily exercise activities.
	Personalize the rehabilitation program	Develop an individually tailored exercise program.
		Tailor the instruction and program to make the task understandable.
	Manage unpleasant sensations	Identifying the challenges of postoperative rehabilitation through discussion.
		Use prescribed medications or heat/ice treatment to relieve or decrease pain.
Vicarious experience	Sharing cases	Share previous success stories to build confidence and motivate patients.
	Problem-solving method	Identify obstacles to participating in the rehabilitation program.
		Use the problem-solving method to address perceived obstacles to participation in rehabilitation programs.
Verbal persuasion	Persuasion	Describe the benefits of physical activities.
		Behavioral contract.
		Regular contact with patients via phone.
	Encouragement and compliments	Assert that participants can self-manage.
		Provide positive verbal feedback on their efforts.
		Give verbal encouragement and compliments.
		Motivation interviewing.
		Reinforce participants’ past and present successes or accomplishments.
		Family involvement.
		Digital activity coaching system.
Physiological and affective states	Avoid negative emotional stimulation	Assess patients’ expressions of anxiety and depression.
	Help to seek support	Telephone-assisted counseling.
		Identify individual barriers and resources for performing the exercise plan.
		Provide strategies for dealing with the identified barriers and coping in the future.

## Data Availability

The data that support the findings of this study are available from the corresponding author upon reasonable request.

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
