# Peer review of "Health Professionals’ Motivational Strategies to Enhance Adherence in the Rehabilitation of People with Lower Limb Fractures: Scoping Review"

_ijerph, 2023, doi:10.3390/ijerph20227050_

Round 1
Reviewer 1 Report
Comments and Suggestions for Authors
This is an important review question. However, I was not convinced that all of the relevant literature had been found as my previous and current work, which is about enhancing rehabilitation following hip fracture, was not described. It was not clear how "the motivational strategies used by health professionals" was assessed and the full search criteria were not described in an appendix. No exclusion criteria were described. In the PRISMA flow diagram, the reasons for the excluded studies are not stated. Common themes that emerged from the extracted data were classified according to Bandura's four sources of self-efficacy. Were any themes identified that fell outside of this self-efficacy model?
Author Response
We want to thank the reviewer for the thoughtful comments and efforts toward improving and building on the manuscript. We have incorporated changes to reflect all the suggestions provided. We have highlighted these changes within the manuscript.
Comment:
This is an important review question. However, I was not convinced that all of the relevant literature had been found as my previous and current work, which is about enhancing rehabilitation following hip fracture, was not described.
Answer to Reviewer
Your valid concerns regarding the limitations of our scoping review are appreciated, and we want to address them. Acknowledging that any review may only encompass part of the existing literature due to various factors influencing study selection is essential. We narrowed our search to five specific databases and used specific descriptors as part of our search strategy. We acknowledge and understand that due to these choices and language constraints, some relevant articles may have been excluded. These limitations are inherent to the scoping review process, and to provide clarity and transparency, we have now elaborated on them in our limitations section. In addition, it's important to note that this scoping review represents the initial phase of a broader research endeavor aimed at developing a care pathway to empower healthcare professionals in motivating patients with lower limb fractures to adhere to rehabilitation plans. This review's findings serve as the foundational framework for an international study involving various stakeholders through a Delphi panel approach.
Comment:
It was not clear how "the motivational strategies used by health professionals" was assessed and the full search criteria were not described in an appendix. No exclusion criteria were described.
Answer to Reviewer
We have now introduced a table (table1) in the text outlining the search criteria used by researchers.
Comment:
In the PRISMA flow diagram, the reasons for the excluded studies are not stated.
Answer to Reviewer
We have revised the PRISMA flow diagram to address the suggestion. In the updated version, we now include reasons for excluding studies.
Comment:
Common themes that emerged from the extracted data were classified according to Bandura's four sources of self-efficacy. Were any themes identified that fell outside of this self-efficacy model?
Answer to Reviewer
All the themes identified from the extracted data were successfully classified within Bandura's four sources of self-efficacy. No themes fell outside of this self-efficacy model, indicating that the interventions reviewed in this study align with the framework proposed by Bandura.
Reviewer 2 Report
Comments and Suggestions for Authors
Thank you for giving this opportunity to review the manuscript titled “Health professionals’ motivational strategies to enhance adherence in the rehabilitation of people with lower limb fractures: Scoping review”. The authors did a good job in presenting their ideas. Some suggestions to improve the paper are:
1) Can the manuscript be revised to enhance information and language clarity? Please do so where appropriate.
2) Can the authors provide a more thorough description of the methodology used in the study? This would make it stronger in details.
3) Are the search strategies used to identify relevant studies adequately explained? Support your ideas with citations/references.
4) Suggest to add more details about the process of data extraction for better understanding.
5) Please discuss the implications of the scoping review findings in relation to practice, policy, or future research.
6) Please ensure the consistency in the references list, following the journal guidelines.
Author Response
We want to thank the reviewer for the thoughtful comments and efforts toward improving and building on the manuscript. We have incorporated changes to reflect all the suggestions provided. We have highlighted these changes within the manuscript.
Comment:
Thank you for giving this opportunity to review the manuscript titled “Health professionals’ motivational strategies to enhance adherence in the rehabilitation of people with lower limb fractures: Scoping review”. The authors did a good job in presenting their ideas. Some suggestions to improve the paper are:
1) Can the manuscript be revised to enhance information and language clarity? Please do so where appropriate.
Answer to Reviewer
We have revised the text to enhance its language clarity and coherence.
Comment:
2) Can the authors provide a more thorough description of the methodology used in the study? This would make it stronger in details.
3) Are the search strategies used to identify relevant studies adequately explained? Support your ideas with citations/references.
4) Suggest to add more details about the process of data extraction for better understanding.
Answer to Reviewer
In response to the reviewer's input, we have expanded the 'Methodology' section to offer a more comprehensive account of the research methodology utilized in this review.
Comment:
5) Please discuss the implications of the scoping review findings in relation to practice, policy, or future research.
Answer to Reviewer
We have considered the reviewer's feedback and have enhanced the 'Strengths and Limitations' section.
It now reads:
This review has merits by identifying different strategies to motivate patients with lower limb fractures to adhere to rehabilitation programs. In addition, to serve as a platform for helping other professionals to motivate patients, the findings from this review can have implications for various stakeholders.
The diverse strategies uncovered in this review lay the groundwork for future research endeavors. They provide a rich resource for researchers to delve deeper into the effectiveness and nuances of these motivational techniques. By building upon the foundation established in this review, researchers can explore innovative approaches, assess long-term outcomes, and refine existing strategies.
The strategies unveiled can readily translate into practical interventions for healthcare professionals. The review is a resource to guide health professionals in their daily practice. It equips them with a toolbox of effective motivational techniques that they can integrate into their patient care routines. This, in turn, can enhance patient engagement and adherence, ultimately improving the quality of care.
One of the notable implications is the potential development of specialized care pathways. The findings can serve as a foundational structure to design a care pathway explicitly aimed at motivating patients with lower limb fractures to adhere to rehabilitation programs. Such a pathway can outline a systematic and evidence-based approach to treatment, addressing the unique needs and challenges of this patient population.
The findings of this review not only provide immediate value but also have far-reaching implications. They serve as a foundation for future research, inspire the development of tailored care pathways, empower healthcare professionals, and, most importantly, contribute to the well-being and recovery of patients with lower limb fractures.
Comment:
6) Please ensure the consistency in the references list, following the journal guidelines.
Answer to Reviewer
We appreciate the reviewer's attention to detail and have taken steps to ensure consistency in the references list, aligning it with the specific guidelines of the journal.
Reviewer 3 Report
Comments and Suggestions for Authors
Thank you for the opportunity to revise this interesting scoping review study. Although the authors should be congratulated on the effort.
The manuscript is clear, relevant to the field, and is presented in a well-structured manner. The statements and conclusions are drawn coherently and are well-supported by the cited references. The results presented in this review significantly contribute to the current understanding of motivational strategies employed by health professionals in the rehabilitation of individuals with lower limb fractures. The comprehensive and well-organized nature of this research makes a valuable addition to the existing body of knowledge in this area, offering meaningful insights for both researchers and practitioners in the field.
The research question is clear and sufficient search bases have been used.
My decision is to accept with minor revisions. Here are some aspects to enhance in certain sections Publication Year Range Justification: The decision to include studies published between 2003 and 2023 should be justified. Explain the rationale behind this specific time frame, as it impacts the comprehensiveness of your review.
Quality Assessment: Consider incorporating a quality assessment of the included studies.
Assessing the quality of studies can add depth to the scoping review and help readers evaluate the strength of the evidence.
Inter-Rater Reliability: It's essential to report the level of agreement between the two researchers who independently screened studies and resolved discrepancies, as this ensures transparency in the study selection process.
Author Response
We want to thank the reviewer for the thoughtful comments and efforts toward improving and building on the manuscript. We have incorporated changes to reflect all the suggestions provided. We have highlighted these changes within the manuscript.
Comment:
Thank you for the opportunity to revise this interesting scoping review study. Although the authors should be congratulated on the effort.
The manuscript is clear, relevant to the field, and is presented in a well-structured manner. The statements and conclusions are drawn coherently and are well-supported by the cited references. The results presented in this review significantly contribute to the current understanding of motivational strategies employed by health professionals in the rehabilitation of individuals with lower limb fractures. The comprehensive and well-organized nature of this research makes a valuable addition to the existing body of knowledge in this area, offering meaningful insights for both researchers and practitioners in the field.
The research question is clear and sufficient search bases have been used.
My decision is to accept with minor revisions. Here are some aspects to enhance in certain sections Publication Year Range Justification: The decision to include studies published between 2003 and 2023 should be justified. Explain the rationale behind this specific time frame, as it impacts the comprehensiveness of your review.
Answer to Reviewer
We selected this time frame to strike a balance between depth and breadth. It allowed us to encompass the most contemporary and relevant research in the field while maintaining feasibility, considering practical limitations in resources and time.
Extending the time frame to 2003 ensured we had a substantial body of literature to review, providing a broad range of studies and interventions. This approach enables a comprehensive review that adequately represents the diversity of motivational strategies in this domain, which was our primary aim.
This time frame effectively addresses our review's objectives while optimizing resource utilization and time management.
To provide clarity, we have included a more detailed explanation in the article outlining the reasoning behind our time frame selection.
Comment:
Quality Assessment: Consider incorporating a quality assessment of the included studies.
Assessing the quality of studies can add depth to the scoping review and help readers evaluate the strength of the evidence.
Answer to Reviewer
The framework proposed by Arksey and O'Malley stated that ‘quality assessment does not form part of the scoping review. Search strategies for scoping reviews are determined by the time available and the specific scope of the topic area. They do not usually involve quality assessment (whereas studies within systematic reviews are critically appraised for quality and risk of bias).
The primary focus is goal is to map the available literature on a specific topic, identify knowledge gaps, and provide a broad overview of the field of study. However, we acknowledge this limitation and have highlighted it in the limitations section.
Comment:
Inter-Rater Reliability: It's essential to report the level of agreement between the two researchers who independently screened studies and resolved discrepancies, as this ensures transparency in the study selection process.
Answer to Reviewer
The inter-rater reliability at the end of the screening process was 94.7%, and at the eligibility process, 100%. We have added this information to the manuscript.
Round 2
Reviewer 1 Report
Comments and Suggestions for Authors
Sorry, but you have not addressed my concerns about your search strategy. Replying that systematic review searches are incomplete by their very nature does not address the point. This is a scoping review, and I would expect you to identify relevant papers, however you have not identified the FEMuR publications:
(for example, Williams NH, Roberts JL, Din NU, Charles JM, Totton N, Williams M, et al A multidisciplinary rehabilitation package following hip fracture: Fracture in the Elderly Multidisciplinary Rehabilitation (FEMuR). Health Technol Assess 2017; 21(44).
or:
Williams NH, Roberts JL, Din NU, Totton N, Charles JM, Hawkes CA, et al. Fracture in the Elderly Multidisciplinary Rehabilitation (FEMuR): A phase II randomised feasibility study of a multidisciplinary rehabilitation package following hip fracture. BMJ Open 2016, 6, e012422.)
or explain to me why they were excluded.
Responses to Reviewer
We thank the reviewer for highlighting the specific articles, particularly the works by Williams et al. (2017) and Williams et al. (2016). It is important to emphasize that the articles mentioned above will bolster our findings and be considered in the next stage of our research, which involves an international study with various stakeholders through a Delphi panel approach. These references will undoubtedly play a significant role in our in-depth exploration of the subject. However, it's important to note that these articles were not part of the initial list of articles identified during the review's identification phase. As such, it wasn´t included.